# Prevalence and associated factors of needle stick and sharps injuries among healthcare workers in northwestern Ethiopia

Zemene Berhan[1], Asmamaw Malede[2], Adinew Gizeyatu[2], Tadesse Sisay[2], Mistir Lingerew[2], Helmut Kloos[3], Mengesha Dagne[2], Mesfin Gebrehiwot[2], Gebremariam Ketema[4], Kassahun Bogale[4], Betelhiem Eneyew[2], Seada Hassen[2], Tarikuwa Natnael[2], Mohammed Yenuss[2], Leykun Berhanu[2], Masresha Abebe[2], Gete Berihun[2], Birhanu Wagaye[5], Kebede Faris[2], Awoke Keleb[2], Ayechew Ademas[2], Akalu Melketsadik Woldeyohanes[2], Alelgne Feleke[2], Tilaye Matebe Yayeh[6], Muluken Genetu Chanie[7], Amare Muche[8], Reta Dewau[8], Zinabu Fentaw[8], Wolde Melese Ayele[8], Wondwosen Mebratu[8], Bezawit Adane[8], Tesfaye Birhane Tegegne[9], Elsabeth Addisu[9], Mastewal Arefaynie[9], Melaku Yalew[9], Yitayish Damtie[9], Bereket Kefale[9], Zinet Abegaz Asfaw[9], Atsedemariam Andualem[10], Belachew Tegegne[10], Emaway Belay[11], Metadel Adane[2]*

1 Quality Improvement Unit, Finote Selam General Hospital, Finote Selam, Ethiopia, 2 Department of Environmental Health, College of Medicine and Health Sciences, Wollo University, Dessie, Ethiopia, 3 Department of Epidemiology and Biostatistics, University of California, San Francisco, California, United States of America, 4 Department of Pharmacy, College of Medicine and Health Sciences, Wollo University, Dessie, Ethiopia, 5 Department of Public Health Nutrition, College of Medicine and Health Sciences, Wollo University, Dessie, Ethiopia, 6 Department of Statistics, College of Natural Sciences, Wollo University, Dessie, Ethiopia, 7 Department of Health Systems and Policy, College of Medicine and Health Sciences, Wollo University, Dessie, Ethiopia, 8 Department of Epidemiology and Biostatistics, School of Public Health, College of Medicine and Health Sciences, Wollo University, Dessie, Ethiopia, 9 Department of Reproductive and Family Health, School of Public Health, College of Medicine and Health Sciences, Wollo University, Dessie, Ethiopia, 10 Department of Nursing, School of Nursing and Midwifery, College of Medicine and Health Sciences, Wollo University, Dessie, Ethiopia, 11 Department of Public Health, College of Health Sciences, Debre Tabor University, Debre Tabor, Ethiopia

☯ These authors contributed equally to this work.
* metadel.adane2@gmail.com

**Data Availability Statement:** All relevant data are within the manuscript and its Supporting information files.

## Abstract

### Background

Needle stick and sharp injuries (NSSIs) are a common problem among healthcare workers (HCWs). Although the factors related to NSSIs for HCWs are well documented by several studies in Ethiopia, no evidence has been reported about the magnitude of and factors related to NSSIs in hospitals in northwestern Ethiopia.

### Methods

An institution-based cross-sectional study was carried out from January to March 2019 among 318 HCWs in three randomly-selected hospitals of the eight hospitals found in South Gondar Zone. Sample sizes were proportionally allocated to professional categories. Study participants were selected by systematic random sampling methods using the monthly salary payroll for each profession as the sampling frame. Data were collected using a self-

**Funding:** Amhara Regional Health Bureau funded this study. The funders had no role in study design, data collection and analysis, decision to publish, or preparation of the manuscript.

**Competing interests:** The authors have declared that no competing interests exist.

**Abbreviations:** AOR, adjusted odds ratio; CI, confidence interval; COR, crude odds ratio; HCWs, healthcare workers; IPPS, infection prevention and patient safety; NSSIs, needle stick and sharps injuries; PPE, personal protective equipment.

administered questionnaire. The outcome of this study was the presence (injured) or absence of NSSIs during the 12 months prior to data collection. A binary logistic regression model with 95% confidence interval (CI) was used for data analysis. Variables from the bivariable analysis with a $p$-value $\leq$ 0.25 were retained into the multivariable analysis. From the multivariable analysis, variables with a $p$-value less than 0.05 was declared as factors significantly associated with NSSIs.

## Main findings

The prevalence of NSSIs was 29.5% (95% CI: 24.2–35.5%) during the 12 months prior to the survey. Of these, 46.0% reported that their injuries were moderate, superficial (33.3%) or severe (20.7%). About 41.4% of the injuries were caused by a suture needle. Factors significantly associated with NSSIs were occupation as a nurse (adjusted odds ratio [AOR] = 2.65, 95% CI: 1.18–4.26), disposal of sharp materials in places other than in safety boxes (AOR = 3.93, 95% CI: 2.10–5.35), recapping of needles (AOR = 2.27, 95% CI: 1.13–4.56), and feeling sleepy at work (AOR = 2.24, 95% CI: 1.14–4.41).

## Conclusion

This study showed that almost one-third of HCWs had sustained NSSIs, a proportion that is high. Factors significantly associated with NSSIs were occupation as a nurse, habit of needle recapping, disposal of sharp materials in places other than in safety boxes and feeling sleepy at work. Observing proper and regular universal precautions for nurses during daily clinical activities and providing safety boxes for the disposal of sharp materials, practicing mechanical needle recapping and preventing sleepiness by reducing work overload among HCWs may reduce the incidence of NSSIs.

## Background

Healthcare workers (HCWs) are at risk of acquiring life-threatening blood-borne infections through needle stick and sharps injuries (NSSI) in their work place [1]. NSSIs occur during screening, diagnosing, treating, and monitoring patients, and disposal of needles and other sharp materials. HCWs who sustain NSSIs experience psychiatric morbidity such as depression, post-traumatic stress disorder, and adjustment disorder. The consequences of these effects include absenteeism and poor healthcare service delivery [2].

Globally, 86% of occupationally related infections are reportedly due to needle stick injuries [3] and the disease burden caused by percutaneous sharps injuries is approximately 3 million infections per year [2]. The burden of needle sticks and other percutaneous injuries among HCWs in Germany and UK were reported to be 500,000 and 100,000 per year, respectively [4]. HCWs are at risk of acquiring hepatitis B virus (HBV), hepatitis C virus (HCV) and HIV infections by sharps injuries [2, 4, 5]. About 40% of all HBV, 40% of HCV, and 4.4% of HIV/AIDS cases among HCWs are due to NSSIs [2].

In sub-Saharan Africa, many NSSIs are due to overwork and inadequate personal protective equipment (PPE), resulting in multiple injuries per HCW each year [6]. One study revealed that the prevalence of NSSIs among HCWs in sub-Saharan Africa was 32.0% in 2013 [7]. NSSIs have several routes of exposure: for instance in northern Uganda, 5.1% of HIV exposure

was associated with sharp objects [8], and 57% of the nurses and midwives had experienced at least one needle stick injury per year [9]. A study in Kenya's Rift Valley Provincial Hospital reported that 19% of health care workers reported having sustained percutaneous injuries, 7.2% splashes to mucosal membranes, and 25% exposure to blood and other body fluids in the past 12 months. High rates of percutaneous injuries were reported by nurses (50%) during stitching (30%) and in the obstetric and gynecologic department (22%) [10].

In Ethiopia, studies conducted in Addis Ababa and Bale Zone reveal that 66.6% and 39.3% of HCWs had sustained NSSI, respectively [11, 12]. The Federal Ministry of Health of Ethiopia developed guidelines for infection prevention and post-exposure prophylaxis use in 2004, 2005 and 2015 [13]. Their aim was to prevent NSSIs among HCWs through ensuring clean and safe health facilities.

Needle-stick incidents are associated with a number of different work activities, including heavy workload, working in surgical or intensive care units, insufficient work experience, and young age [14]. Although data on the prevalence of NSSIs and associated factors among HCWs exist in many larger urban health facilities in Ethiopia [12, 15–19], these study findings are not comparable due to variations in healthcare delivery, occupations of HCWs, methods of injection, drawing of blood and needle disposal, and the practice of recapping needles [20, 21]. Moreover, no study has been conducted in South Gondar Zone hospitals in northwestern Ethiopia to identify the prevalence of NSSI and associated factors among the area's HCWs, which hinders appropriate actions to prevent them. This study was designed to provide such local evidence.

## Methods

### Study setting

This study was conducted in South Gondar Zone hospitals in northwestern Ethiopia. South Gondar Zone, one of the 13 zones in Amhara Region, is divided into 18 districts. Its capital city is Debre Tabor, which is about 600 km north of Addis Ababa and about 110 km east of Bahir Bahir Dar. The study included one general government hospital (Debre Tabor general hospital) and two of the seven district hospitals in South Gondar Zone, Amhara National Regional State.

Debre Tabor general hospital is found in Debre Tabor Town and the seven district primary government hospitals are in Addis Zemen, Mekane Eyesus, Andabet, Ebnat, Nefas Mewcha, Arb Gebeya, and Smada towns. In addition, there are 98 government health centers and 76 private clinics in South Gondar Zone [22]. After this study conducted, Debre Tabor general hospital was promoted to Debre Tabor Comprehensive Specialized Hospital and Tach Gayint Primary Hospital was renamed by Dr. Ambachew Mekonen Memorial Primary Hospital. Hereafter, we used the new names for consistency with the future studies.

### Study design and source population

An institution-based cross-sectional study was conducted from January to March 2019. The source population of this study consisted of all HCWs working in eight South Gondar Zone government hospitals. The study populations were HCWs working in the three randomly selected hospitals for this study.

### Inclusion and exclusion criteria

The HCWs participating in this study included nurses, midwives, laboratory technicians, health officers, medical doctors (general practitioners, gynecologists/obstetricians,

anesthesiologists, internists, pediatricians, surgeons, and ophthalmologists), dentists, cleaners and laundry staff. However, pharmacists and environmental health professionals were excluded in this study because they are less vulnerable for NSSIs.

## Sample size determination and sampling procedures

Sample size was determined using a single population proportion ($n = \frac{(z_{a/2})^2 * p(1-p)}{d^2}$) formula [23] with an assumption of $Z_{\alpha/2}$ at 95% confidence interval is 1.96; $d$ is degree of error of 5%; and proportion ($p$) of NSSIs among HCWs of 32.8% was taken from a study done in Debre Birhan hospitals in Amhara Region [19]. We used a design effect of 1.5 since we employed a multi-stage sampling method, giving a calculated sample size of 508. Furthermore, since the source population in South Gondar Zone hospitals was less than 10,000, we used a sample size correction formula of $n/[1+ (n$-1$)/N]$ [23], where $n$ is the initial calculated sample size (508) and $N$ is the source population (749). Then, the sample size became 303. Finally, a 5% non-response rate was added and a final sample size of 318 was obtained.

A two-stage sampling method was employed. During the first stage, Debre Tabor Comprehensive Specialized hospital, and Mekane Eyesus and Dr. Ambachew Mekonen Memorial Primary Hospitals were selected using the lottery method from the eight hospitals found in South Gondar. A total of 442 HCWs worked in the three hospitals, 320 of them in Debre Tabor Comprehensive Specialized Hospital, 71 in Mekane Eyesus Primary Hospital and 51 in Dr. Ambachew Mekonen Memorial Primary Hospital. Based on the number of HCWs in each hospital, the sample size for this study was proportionally allocated to the three selected hospitals. Similarly, the sample size of each category of profession (nurses, medical doctors, laboratory technicians, health officers, cleaners and laundry workers) was proportionally allocated.

At the second stage, the participating HCWs were selected by using the monthly salary payroll as the sampling frame. Thus, a separate sampling frame was prepared for each profession based on the monthly salary payroll. Then, the study participants in each profession who adhered to the inclusion criteria were selected using a systematic random sampling technique.

## Operational definitions

**Healthcare Workers (HCWs).**   In this study, HCWs were nurses, midwives, laboratory technicians, health officers, general practitioners, gynecologists/obstetricians, anesthesiologists, internists, pediatricians, surgeons, and ophthalmologists, dentists, cleaners and laundry staff whose activities involved contact with needles and other sharps during the course of their work in a healthcare facility [2].

**Medical sharps.**   Any object used in the healthcare setting that can penetrate the skin, including suture needles, hypodermic needles, disposable needles, blood sugar lances, surgical scalpels, trocar puncture needles, vacuum tube blood collection needles, broken vials or ampules, razors, scissors, scalpels, lancets, retractors, broken capillary tubes, and glassware [2].

**Needle Stick and Sharps Injury (NSSI).**   The outcome variable of this study is the presence or absence of NSSI during the 12 months prior to data collection. The presence of NSSIs was measured by self-reporting the penetration of the skin by needles or other sharp objects that had been in contact with blood, tissue, or a body fluid before the exposure [2].

**Needle Stick and Sharps Injury (NSSI) types.**   NSSI can be classified as moderate, severe and superficial. The moderate category includes puncturing of the skin involving some bleeding; severe NSSIs include deep sticks/cut or profuse bleeding and the superficial group injuries caused by sharps that resulted in little or no bleeding [24].

**Prevalence of NSSI.**   The ratio of the number of HCWs who sustained NSSIs to the total number of HCWs during the 12 months prior to data collection multiplied by 100.

**Personal Protective Equipment (PPE).** Equipment designed to protect workers from workplace injuries or illnesses resulting from contact with blood, body fluid, and radiological, physical, mechanical, or other workplace hazards. This includes a variety of devices and garments, such as masks, gloves and eye goggles [25].

**Recapping.** The act of replacing a protective sheath on a needle [26].

**Universal precautions.** The practice of standard set of guidelines by healthcare workers to avoid contact with patients' bodily fluids for the prevention of the transmission of blood-borne pathogens [27].

## Data collection tools, data collection and quality assurance

Data were collected using a structured, self-administered questionnaire. The questionnaire was prepared after reviewing similar studies on NSSIs [11, 28]. The questionnaire was first prepared in English, then translated to Amharic (local language), and then retranslated back to English to check for consistency. The instrument elicited information on socio-demographic characteristics of respondents and organization-related, skill-related, and behavior-related factors. Three data collectors with BSc degrees in nursing were recruited from Debre Tabor Town and were trained for one day on the study instrument and data collection procedures. Then, one data collector was assigned to each hospital to collect data using a self-administered questionnaire from the participating HCWs who had contact with sharps and needle instruments during the course of their work in the 12-month recall period preceding the survey.

Data quality was assured during questionnaire design and data collection, entry, and analysis. The questions were objective, non-leading, logically sequenced, and free of scientific jargon. To ensure the validity of the data collection tool, inter-observer reliability was ensured by providing clear definitions of measured variables, and events to be recorded. We re-self-administered 5% of the study participants to check reliability of the information entered at different times about the same study participant. Furthermore, to ensure the content of the survey tool was valid, the questionnaire was pretested in a 10% sample of the study's sample of HCWs in nearby hospital (Lay Gayint Hospital). Based on the pre-test responses, questions were revised as necessary.

The principal investigator provided one day's training to data collectors, and then reviewed the collected data each day, returning incomplete questionnaires to data collectors who in turn contacted the study participants the same day. In order to reduce social desirability bias in answers, a self-administered survey and closed-ended questions were used. In order to verify the accuracy of data entries, two generic data verification strategies were employed as described in another study [29]. As the first step, randomly selected 10% of the questionnaires were thoroughly checked. Following this, descriptive statistics, results from cross-tabulations, and frequency distributions were examined before performing statistical analysis.

## Data management and analysis

The collected data were entered into EpiData version 3.1 (EpiData Association, Odense, Denmark) and exported to Statistical Package for the Social Sciences (SPSS) version 24.0 software (IBM Corp., Armonk, N.Y., USA) for data cleaning and analysis. Descriptive statistics such as frequencies and percentages were calculated to examine the overall distribution of the variables. Multicollinearity was checked using standard error of the coefficient with a cut-off point of 2 [30].

A binary logistic regression model was used to examine the association between independent variables and NSSI. Independent variables having $p$-value $\leq 0.25$ from the bi-variable analysis were retained into multivariable analysis. Then, in the multivariable analysis,

$p$-value $< 0.05$ and AOR (adjusted odds ratio) with 95% CI were used to measure associations; variables with $p$-value $< 0.05$ were declared as statistically significant and associated factors of NSSI. Model fitness was checked using the Hosmer and Lemeshow test [30] to conduct logistic regression analysis when the model is fit at $p$-value $> 0.05$.

### Ethical consideration

This study adhered to the ethical principles of the Declaration of Helsinki [31] and the principles that govern medical research involving human subjects [32]. Thus, ethical clearance was obtained from the Ethical Review Committee of Wollo University, College of Medicine and Health Sciences. The study participants were informed of the purpose of the study before asking for their written consent for participation. The respondents' right to refuse or withdraw from participation in the study was fully maintained and the information provided by each respondent was kept confidential through the use of codes rather than names. Study participants who were with NSSI during data collection and who had not recovered from their injuries were advised to get treatment.

## Results

### Socio-demographic characteristics of healthcare workers

Of the 318 HCWs selected for study, 295 (92.8%) participated. They included 124 (42.0%) nurses, 25 (8.5%) midwives, 21 (7.1%) laboratory technicians, 26 (8.8%) medical doctors (general practitioners), 52 (17.6%) cleaners and laundry workers and 45 (15.3%) other healthcare professionals. Nearly half 140 (47.5%) of the respondents were between 25–30 years old, 148 (50.2%) were females and 181 (61.4%) were unmarried. Seven out of 10 (69.8%) HCWs had less than five years of work experience (Table 1).

### Organization-related characteristics

About 295 HCWs ($n = 259$, 87.8%) reported working 8 or fewer hours per day. More than two-thirds (68.1%) did not work night shifts. One hundred eighty-five (62.7%) HCWs knew that a safety protocol was in place, but most (71.9%) did not know that universal precautions posters were posted in their institutions. About 61.4% of the respondents disposed of needles and other sharp materials in safety boxes and 178 (60.3%) had boxes for sharps in their work rooms (Table 2).

### Behavioral characteristics

Almost half 143 (48.5%) of the HCWs reported that they habitually recapped needles, 72 (24.4%) reported feeling sleepy at work, 53 (18.0%) drank alcohol, 8 (2.7%) chewed *chat* (*Catha edulis*), and 14 (4.7%) smoked cigarettes occasionally. Two hundred-seventy (91.5%) of the HCWs knew about the risk of disease transmission through NSSIs (Table 3).

### Skill-related characteristics

Nearly three-quarters ($n = 213$, 72.2%) of the HCWs were not trained about infection prevention and 208 (70.5%) received no training about patient safety and injection safety. About 259 (87.8%) had no access to information about NSSIs and 277 (93.9%) had no knowledge of how to prevent NSSIs (Table 4).

**Table 1. Socio-demographic characteristics and bi-variable analysis with NSSIs among healthcare workers in South Gondar Zone hospitals, January to March 2019.**

| Variable | Category | Frequency | Injury | | COR (95% CI) | P-value |
|---|---|---|---|---|---|---|
| | | *n* (%) | Yes | No | | |
| | | | *n (%)* | *n (%)* | | |
| HCW's age (years) | < 25 | 70 (23.7) | 22 (25.3) | 48 (23.1) | 0.39(0.12–1.30) | 0.127 |
| | 25–30 | 140 (47.5) | 38 (4.4) | 102 (49.0) | 0.32(0.10–1.01) | 0.052 |
| | | 72 (24.4) | 20 (23.0) | 52 (25.0) | 0.33(0.99–1.10) | 0.071 |
| | >40 | 13 (4.4) | 7 (2.4) | 6 (2.9) | 1 | |
| HCW's sex | Male | 147 (49.8) | 45 (51.7) | 102 (49.0) | 1.11(0.67–1.83) | 0.67 |
| | Female | 148 (50.2) | 42 (48.3) | 106 (51.0) | 1 | |
| HCW's marital status | Married | 114 (38.6) | 36 (41.4) | 78 (37.5) | 1.17(0.70–1.96) | 0.533 |
| | Unmarried | 181 (61.4) | 51 (58.6) | 130 (44.1) | 1 | |
| HCW's profession | Nurse | 124 (42.0) | 51 (58.6) | 73 (24.7) | 3.21(1.82–5.92) | 0.030 |
| | Medical doctor (general practitioner) | 26 (8.8) | 6 (6.9) | 22 (7.5) | 0.86(0.23–3.12) | 0.823 |
| | Laboratory technician | 21 (7.1) | 6 (6.9) | 15 (5.1) | 1.26(0.33–4.73) | 0.725 |
| | Cleaner or laundry worker | 52 (17.6) | 10 (3.4) | 42 (20.2) | 0.75(0.23–2.37) | 0.534 |
| | Midwife | 25 (8.5) | 6 (6.9) | 19 (9.1) | 1 | |
| | Other [a] | 45 (15.3) | 8 (9.2) | 37 (17.8) | 0.68(0.20–2.26) | |
| Education status | Diploma or lower[£] | 111 (37.6) | 26 (29.9) | 85 (40.9) | 0.85(0.36–1.99) | 0.713 |
| | BSc | 136 (46.4) | 46 (52.9) | 90 (43.3) | 1.60(0.72–3.58) | 0.242 |
| | MSc | 10 (3.1) | 5 (5.7) | 5 (2.4) | 0.35(0.09–3.16) | 0.357 |
| | MD | 38 (12.9) | 10 (11.5) | 28 (13.5) | 1 | |
| Work experience (years) | < 5 | 206 (69.8) | 56 (64.4) | 150 (72.1) | 0.51(0.22–1.17) | 0.114 |
| | 5–10 | 63 (21.4) | 20 (23.0) | 43 (20.7) | 0.63(0.24–1.62) | 0.345 |
| | >10 | 26 (8.8) | 11 (12.6) | 15 (7.2) | 1 | |
| Monthly salary (USD)[¥] | < 53.80 | 55 (18.6) | 10 (11.5) | 45 (7.2) | 0.54(0.24–1.21) | 0.146 |
| | 53.80–107.60 | 47 (15.9) | 18 (20.7) | 29 (13.9) | 1.50(0.72–3.14) | 0.279 |
| | 10.60–161.41 | 97 (32.9) | 31 (35.6) | 66 (31.7) | 1.14(0.62–2.10) | 0.677 |
| | >161.41 | 96 (32.5) | 28 (32.2) | 68 (32.7) | 1 | |

1, reference category; CI, confidence interval; COR, crude odds ratio.

BSc, Bachelor of Science; MSc, Master of Science, MD, Medical Doctor including general practitioners, gynecologists/obstetricians, anesthesiologists, internists, pediatricians, surgeons, and ophthalmologists.

[a]Health officer, dentist, gynae/obstetrician, anesthesiologist, internist, pediatrician, surgeon, and ophthalmologist.

[¥]1 $ USD (United States Dollars) = 27.88 (ETB [Ethiopian birr]) during January to March, 2019.

[£]Education status of *lower* indicates that healthcare workers are included who do not have a diploma, such as cleaners and laundry workers.

## Prevalence of Needle Stick and Sharps Injuries (NSSIs)

The overall prevalence of NSSIs among HCWs was 29.5% with a 95% CI (24.2–35.5%) during the 12 months prior to the survey. Among the 87 injured respondents, 40 (46.0%) reported that their injuries were moderate, 29 (33.3%) reported superficial injuries, and 18 (20.7%) reported severe injuries. Sixty-eight (78.2%) had sustained injuries only one time in the previous 12 months and 8.0% recalled three or more injuries. Thirty-six (41.4%) of the injuries were caused by suture needles and 27.6% by disposable syringes (Table 5).

## Factors associated with needle stick and sharps injuries

In the bi-variable logistic regression analyses of the variables presented in Tables 1–4, the following were candidates for multivariable regression with *p*-value ≤ 0.25: occupation, level of

**Table 2. Organization-related characteristics and bi-variable analysis with NSSIs among healthcare workers in South Gondar Zone hospitals, January to March 2019.**

| Variable | Category | Frequency | Injury | | COR (95% CI) | P-value |
|---|---|---|---|---|---|---|
| | | n (%) | Yes | No | | |
| | | | n (%) | n (%) | | |
| No. hours worked per day | >8 | 36 (12.2) | 11 (12.6) | 25 (12.0) | 1.05(0.49–2.26) | 0.881 |
| | ≤ 8 | 259 (87.8) | 76 (87.4) | 183 (88.0) | 1 | |
| HCWs working a night shift occasionally | No | 201 (68.1) | 61 (70.1) | 140 (67.3) | 1.14(0.66–1.96) | 0.633 |
| | Yes | 94 (31.9) | 26 (29.9) | 68 (32.7) | 1 | |
| Universal precautions poster posted in your working area | No | 212 (71.9) | 60 (69.0) | 152 (73.1) | 0.82(0.47–1.41) | 0.472 |
| | Yes | 83 (28.1) | 27 (31.0) | 56 (26.9) | 1 | |
| Presence of safety protocols | No | 110 (37.3) | 29 (33.3) | 81 (38.9) | 0.78(0.46–1.32) | 0.369 |
| | Yes | 185 (62.7) | 58 (66.7) | 127 (61.1) | 1 | |
| Presence of IPPS committee | No | 66 (22.4) | 21 (24.1) | 45 (21.6) | 1.15(0.63–2.08) | 0.638 |
| | Yes | 229 (77.6) | 66 (75.9) | 163 (78.4) | 1 | |
| Received HBV prophylaxis** | Yes | 186 (63.1) | 64 (75.6) | 122 (58.7) | 1.96(1.13–3.40) | 0.016 |
| | No | 109 (36.9) | 23 (26.4) | 86 (41.3) | 1 | |
| Method of sharps disposal | Not using safety box | 114 38.6) | 54 (62.1) | 60 (28.8) | 4.03(3.38–6.83) | 0.001 |
| | Using safety box | 181 (61.4) | 33 (37.9) | 148 (71.2) | 1 | |
| Work department | Surgical ward | 40 (13.6) | 12 (13.8) | 28 (13.5) | 1 | |
| | Medical ward | 43 (14.6) | 14 (16.1) | 29 (13.9) | 1.12(0.44–2.85) | 0.803 |
| | Gynecology/obstetrics ward | 31 (10.5) | 19 (21.8) | 22 (10.6) | 2.01(0.80–5.02) | 0.139 |
| | Operating room | 36 (12.2) | 9 (10.3) | 27 (13.0) | 0.77(0.28–2.14) | 0.622 |
| | Pediatric ward | 24 (8.1) | 6 (6.9) | 18 (8.7) | 0.77(0.25–2.44) | 0.665 |
| | Emergency ward | 30 (10.2) | 7 (8.0) | 23 (11.1) | 0.71(0.24–2.09) | 0.531 |
| | Outpatient ward | 39 (13.2) | 10 (11.5) | 29 (13.9) | 0.80(0.30–2.15) | 0.667 |
| | Other * | 42 (14.2) | 10 (11.5) | 32 (15.4) | 0.73(0.27–1.94) | 0.524 |
| Location of sharps container | Within each patient ward | 178 (60.3) | 53 (60.9) | 125 (60.1) | 1 | |
| | Medication cart | 24 (8.1) | 7 (8.1) | 17 (8.2) | 0.97(0.38–2.47) | 0.971 |
| | Within each procedure room | 82 (27.8) | 21 (24.1) | 61 (29.3) | 0.81(0.45–1.46) | 0.498 |
| | Other¥ | 11 (3.7) | 6 (6.9) | 5 (2.4) | 2.83(0.82–9.67) | 0.096 |

1, reference category; HCWs, healthcare workers; HBV, hepatitis B virus; CI, confidence interval; COR, crude odds ratio; IPPS, infection prevention and patient safety

*laboratory, laundry, and neonatal intensive care unit.

¥Outside of the different wards near the door, laboratory room and pharmacy room

**HBV vaccination before working in health care institution

education, monthly income, IPPS training, disposal method of sharp material, HBV vaccination status, habit of needle recapping, feeling sleepy at work, alcohol use, chat use, cigarette use, and injection safety training. After controlling the confounding factors, the following variables were found to be significantly associated with NSSIs ($P$-value < 0.05): occupation, disposal of sharp materials, habit of needle recapping, and feeling sleepy at work.

The analysis shows that the odds of nurses being injured by NSSIs were 2.65 times (AOR = 2.65, 95% CI: 1.18–4.26) higher than for midwives. This study also indicated that HCWs who disposed of sharp materials without safety boxes were 3.93 times (AOR = 3.93, 95% CI: 2.10–5.35) more likely to have NSSIs than those who disposed them in safety boxes. Those workers who reported feeling sleepy at work were 2.24 times (AOR = 2.24, 95% CI: 1.14–4.41) more likely to sustain NSSIs than those who did not feel sleepy at work. Health

**Table 3. Behavioral characteristics and bi-variable analysis with NSSIs among healthcare workers in South Gondar Zone hospitals, January to March 2019.**

| Variable | Category | Frequency | Injury | | COR (95% CI) | P-value |
|---|---|---|---|---|---|---|
| | | n (%) | Yes | No | | |
| | | | n (%) | n (%) | | |
| Practiced needle recapping | Yes | 143 (48.5) | 55 (63.2) | 88 (42.3) | 2.34(1.40–3.92) | 0.001 |
| | No | 152 (51.5) | 32 (36.8) | 120 (57.7) | 1 | |
| Frequency of recapping (N = 143) | All of the time | 52 (36.4) | 23 (41.8) | 29 (32.9) | 1.20(0.56–2.57) | 0.627 |
| | Most of the time | 33 (23.1) | 9 (16.4) | 24 (27.3) | 0.57(0.22–1.44) | 0.235 |
| | Sometime | 58 (40.5) | 23 (41.8) | 35 (39.8) | 1 | |
| Feeling sleepy at work | Yes | 72 (24.4) | 34 (39.1) | 38 (18.3) | 2.87(1.64–5.00) | 0.001 |
| | No | 223 (75.6) | 53 (60.9) | 170 (81.7) | 1 | |
| Awareness of disease transmission by NSSI | Yes | 270 (91.5) | 80 (92.0) | 190 (91.3) | 1 | |
| | No | 25 (8.5) | 7 (8.0) | 18 (8.7) | 1.08(0.43–2.69) | 0.864 |
| Uses PPE | No | 21 (7.1) | 6 (6.9) | 15 (7.2) | 0.95(0.37–2.54) | 0.924 |
| | Yes | 274 (92.9) | 81 (9.3) | 193 (92.8) | 1 | |
| Drinks alcohol occasionally | Yes | 53 (18.0) | 24 (27.6) | 29 (13.9) | 2.35(1.27–4.33) | 0.006 |
| | No | 242 (82.0) | 63 (72.4) | 179 (86.1) | 1 | |
| Chews chat occasionally | Yes | 8 (2.7) | 6 (6.9) | 5 (2.4) | 7.63(1.50–38.58) | 0.014 |
| | No | 287 (97.3) | 81 (93.1) | 203 (97.6) | 1 | |
| Smokes cigarettes occasionally | Yes | 14 (4.7) | 8 (9.2) | 6 (2.9) | 3.40(1.1–10.14) | 0.027 |
| | No | 281 (95.3) | 79 (90.8) | 202 (97.1) | 1 | |

1, reference category; COR, crude odds ratio; CI, confidence interval; NSSI, needle stick and sharps injury; PPE, personal protective equipment.

workers who recapped needles were 2.27 times (AOR = 2.27, 95% CI: 1.13–4.56) more likely to be injured by them than those who did not report this practice (Table 6).

## Discussion

This institution-based cross-sectional study was designed to assess the prevalence of NSSIs and associated factors among HCWs in hospitals of northwestern Ethiopia. We found that the prevalence of NSSIs was 29.5% during the 12 months prior to the survey. Factors significantly

**Table 4. Skill-related characteristics and bi-variable analysis with NSSIs among healthcare workers in South Gondar Zone hospitals, January to March 2019.**

| Variable | Category | Frequency | Injury | | COR (95% CI) | P-value |
|---|---|---|---|---|---|---|
| | | n (%) | Yes | No | | |
| | | | n (%) | n (%) | | |
| Trained in IPPS | No | 213 (72.2) | 55 (63.2) | 158 (76.0) | 0.54(0.31–0.93 | 0.027 |
| | Yes | 82 (27.8) | 32 (36.8) | 50 (24.0) | 1 | |
| Trained in injection safety | No | 208 (70.5) | 56 (64.4) | 152 (73.1) | 0.66(0.39–1.13) | 0.136 |
| | Yes | 87 (29.5) | 31 (35.6) | 56 (26.9) | 1 | |
| Access to information on NSSI | No | 36 (12.2) | 13 (14.9) | 23 (11.1) | 1.41(0.68–2.93) | 0.354 |
| | Yes | 259 (87.8) | 74 (85.1) | 185 (88.9) | 1 | |
| NSSI can be prevented | No | 18 (6.1) | 5 (5.7) | 13 (6.3) | 0.91(0.31–2.64) | 0.869 |
| | Yes | 277 (93.9) | 82 (94.3) | 195 (93.4) | 1 | |

1, reference category; COR, crude odds ratio; CI, confidence interval; IPPS, infection prevention and patient safety; NSSI, needle stick and sharps injury.

**Table 5. Prevalence of NSSIs and characteristics of injured healthcare workers in South Gondar Zone hospitals, January to March 2019.**

| Characteristic | Category | Frequency (*n*) | Percentage (%) |
|---|---|---|---|
| NSSIs (*N* = 295) | Yes | 87 | 29.5 |
| | No | 208 | 70.5 |
| Degree of injury | Severe | 18 | 20.7 |
| | Moderate | 40 | 46.0 |
| | Superficial | 29 | 33.3 |
| Frequency of injuries (*N* = 87) | One time | 68 | 78.2 |
| | Two times | 12 | 13.8 |
| | 3 times or more | 7 | 8.0 |
| Place of care after injury (*N* = 87) | Emergency ward | 48 | 55.2 |
| | IPPS room* | 10 | 11.5 |
| | Outpatient ward | 6 | 6.9 |
| | Did not receive care | 14 | 16.1 |
| | Other | 9 | 10.3 |
| Type of sharp that caused the injuries (*N* = 87) | Suture needle | 36 | 41.4 |
| | Hypodermic needle | 8 | 9.2 |
| | Disposable syringe | 24 | 27.6 |
| | Blood sugar lancet | 6 | 6.9 |
| | Blood collection needle | 6 | 6.9 |
| | Other [¥] | 7 | 8.0 |
| When injury occurred (*N* = 87) | During patient care | 69 | 79.3 |
| | While cleaning room | 5 | 5.7 |
| | During waste disposal | 2 | 2.3 |
| | While walking in the hospital | 7 | 8.1 |
| | While washing clothes | 4 | 4.6 |

[¥]Surgical scalpel blade, phlebotomy needle, broken vial or ampoule repair, scissors, and intravenous catheter stylet.

*Infection prevention and patient safety room (IPPS) as a place of care of injury and where an injured HCW received care for injury.

NSSIs, needle stick and sharp injuries.

**Table 6. Factors associated with NSSIs among healthcare workers from multivariable logistic regression analysis in South Gondar Zone hospitals, January to March 2019.**

| Variable | Category | Injury status | | COR (95% CI) | AOR (95% CI) |
|---|---|---|---|---|---|
| | | Yes (n) | No (n) | | |
| HCW's profession | Nurse | 51 | 73 | 3.21(1.82–5.92) | 2.65(1.18–4.26) |
| | Medical doctor (general practitioner) | 6 | 22 | 0.86(0.23–3.12) | 0.35(0.03–3.54) |
| | Laboratory technician | 6 | 15 | 1.26(0.33–4.73) | 3.75(0.70–19.97) |
| | Cleaner or laundry worker | 10 | 42 | 0.75(0.23–2.37) | 1.28(0.10–16.32) |
| | Midwife | 6 | 19 | 1 | 1 |
| | Other [a] | 8 | 37 | 0.68(0.20–2.26) | 0.43(0.08–2.18) |
| Method of sharps disposal | Not using safety box | 54 | 60 | 4.03(3.38–6.83) | 3.93(2.10–5.35) |
| | Using safety box | 33 | 148 | 1 | 1 |
| Practiced needle recapping | Yes | 55 | 88 | 2.34(1.40–3.92) | 2.27(1.13–4.56) |
| | No | 32 | 120 | 1 | 1 |
| Feeling sleepy at work | Yes | 34 | 38 | 2.87(1.64–5.00) | 2.24(1.14–4.41) |
| | No | 53 | 170 | 1 | 1 |

1, reference category; CI, confidence interval; COR, crude odds ratio; AOR, adjusted odds ratio.

[a]Health officer, dentist, gynae/obstetrician, anesthesiologist, internist, pediatrician, surgeon, and ophthalmologist.

associated with NSSIs were occupation (being a nurse), method of disposal of sharp materials, the practices of needle recapping and feeling sleepy at work.

The prevalence of NSSIs in this study was similar to studies conducted in Tigray Region health facilities (25.9%) [33], and in a Tamil Nadu, India, which reported a one-year prevalence of NSSI of 35.3% among HCWs [1]; in Goa Territory Hospital in India, which reported a 34.8% prevalence [34]; and in sub-Saharan Africa, the average prevalence of NSSI among HCWs to be 32% [7]. In Ethiopia, NSSIs were reported in 26.6% of HCWs in Dire Dawa Town [15] and 32.8% in Debre Berhan Town [19]. In a hospital in Bahir Dar Town, 31.0% of HCWs sustained a NSSI at least once during a 12-month period [18]. These similar rates may be due to the fact that all these facilities are mid-level regional or district hospitals with similar levels of staff training and national IPPS guidelines have been implemented in each hospital.

The prevalence of NSSIs in our study was higher than in studies conducted in Assam, India; Lausanne, Switzerland; and Awi and East Gojjam zones in Ethiopia, where the proportions of injuries during 12-month periods were 21.1%, 9.7%, 18.7% and 22.2%, respectively [24, 35–37]. The possible reasons for these differences might be the lack of adequate sharps disposal sites such as safety boxes and lower adherence to standard precautions. Other reasons might be inadequate training and fewer safety guidelines for the prevention of injuries during patient care in our study hospitals.

The prevalence of NSSIs in our study was also lower than reported by studies conducted in different parts of Ethiopia: 37.1% in Bale Zone hospitals [38]; 39.3% in Jimma Zone hospitals [14]; 35.8% in Hawassa healthcare facilities [16] and 42.8% in Bahir Dar health centers [17]. The difference between the prevalence in our study and those in the other Ethiopian studies may be due to differences in working environments. Furthermore, the functionality of existing IPPS committee, variation in prevention posters displayed in different wards or in the health facility compound and the lack of sufficient safety boxes may have influenced the outcome of these various studies.

The odds of having NSSIs were 2.65 times higher among nurses than among midwives. Consistent with our findings, being a nurse in Poland is also a factor in NSSIs [39]. Several studies also reported high incidence of NSSIs among nurses in Iran [40], in the University of Alexandria teaching hospitals (92.5%) [21], among hospital nurses in South Korea (70.4%) [41], nurses in India (71%) [42] and in Poland (72.6%). This may be because of the high work load of nurses and the high risk of exposure during drug administration and other procedures that require the use of needles and other sharp instruments.

HCWs who disposed of sharp materials without using safety boxes were 3.93 times more likely to sustain NSSIs than those who used safety boxes. Similarly, a study done in London indicated that most NSSIs were caused by disposal of sharp materials without using safety boxes [43]. In a hospital in Iran, the appropriate disposal of used needles nearly eliminated the risk of NSSIs [44]. This discussion indicates that lack of safety boxes, inappropriate and uncontrolled disposal of sharps, and lack of awareness of the risk involved in handling sharps might be largely responsible for NSSI. About 8.1% of NSSI happened while walking in the hospital. This might be due to walking to visit the toilet, during moving from one ward to another, during moving from one location to another during tea break time. This also might have been as a result of lack of cleaning away of sharp materials from the hospital compound. In addition to this, lack of monitoring of the standard infection prevention and control strategies may bring about this horrible situation.

The odds of NSSIs among HCWs who habitually recapped needles were 2.27 higher than for those who did not recap needles. This is consistent with findings from other Ethiopian studies in in Tigray Region, Hawassa, Addis Ababa, Ethiopia [14, 33, 45], Bahir Dar City [18] Jimma Zone, which showed that 37.3% of NSSIs were due to needle recapping, and also

in Bale Zone, where HCWs who practiced needle recapping had a 46% higher risk of NSSI [46].

Furthermore, our findings are also consistent with other country studies; a study conducted in Shiraz, Iran, identified recapping as the major cause of NSSIs [47] and a study in hospitals of Pokhara, Nepal, found recapping to account for 55.1% of NSSIs [48]. A study in a tertiary care hospital in India reported that 63.7% of NSSIs occurred during recapping of needles [49]. In a tertiary care hospital in Assam, recapping was associated with 26.3% of all NSSIs [50].

Workers who reported feeling sleepy at work were 2.24 times more likely to be injured by needles and other sharps than those whose sleep was not disturbed. Feeling sleepy at work may be linked with tiredness and increased vulnerability to NSSIs of HCWs working night shifts. This can be prevented by minimizing clinical activities and adding more HCWs at this time. This finding is consistent with results of a study conducted in East Gojjam Zone [37]. Feeling sleepy at work may also result from stressful psychosocial working conditions [51].

This study had several limitations. The 12-month recall period may have led to underreporting of the prevalence of NSSIs and the circumstances under which they occurred. Moreover, the cross-sectional study design could not establish cause-and-effect relationships due to the retrospective nature of questions on exposure risk. Furthermore, it is difficult to measure feeling sleepy at work by closed-ended yes/no questions due to social desirability bias. We also did not measure the level of substance and alcohol use.

## Implication of the study for practice

This study will have implications for futher strengthening infection prevention and patient safety programs in hospitals to control injuries among healthcare workers. Furthermore, this study will help to prevent diseases due to injuries, including HIV/AIDS and HBV. Controlling injuries at hospitals will also help to ensure healthy workers and thus facilitate the delivery of healthcare services. The findings can guide programmers and managers of hospitals and other stakeholders (government and non-governemntal organziations) to design a mechanism to minimize NSSI and ensure adequate hospital staffing, provision of IPPS training and of necessary safety equipment. The findings may therefore strengthen the promotion and implementation of IPPS programs in hospitals.

## Conclusions

We conclude that almost one-third of the study participants had sustained NSSIs at least once in the previous 12 months. Occupation as a nurse, the habit of needle recapping, feeling sleepy at work and disposing of sharp materials in places other than safety boxes were found to be factors associated with NSSI. To minimize NSSIs, adequate hospital staff recruitment, provision of IPPS training, and provision of necessary safety equipment are recommended. We also recommend promoting and strengthening the implementation of IPPS and strengthening safe hospital committees. This should include provision of safety boxes, mechanical needle recapping devices, health education, and on-the-job training. Health education should emphasize regular use of universal precautions during HCWs' daily activities and dealing with sleepiness at work among HCWs working night and day shifts. We recommend that HCWs working combined day and night shifts work only day or night shifts to prevent sleepiness. Qualitative studies should be triangulated to investigate further factors in NSSIs. Furthermore, a cohort study design incorporating the use of diaries by healthcare workers is recommended to investigate causal relationships and reduce recall bias.

## Supporting information

**S1 Dataset.**
(XLS)

## Acknowledgments

We acknowledged administrators of Debre Tabor Comprehensive Specialized Hospital, Mekane Eyesus and Dr. Ambachew Mekonen Memorial Primary hospitals for their support and permission to conduct this study. The study participant HCWs are also acknowledged for providing relevant information and the data collectors for their assistance. We also thank Lisa Penttila for language editing of the manuscript.

## Author Contributions

**Conceptualization:** Zemene Berhan, Asmamaw Malede, Adinew Gizeyatu, Tadesse Sisay, Metadel Adane.

**Data curation:** Zemene Berhan, Asmamaw Malede, Tadesse Sisay, Tilaye Matebe Yayeh, Metadel Adane.

**Formal analysis:** Zemene Berhan, Asmamaw Malede, Adinew Gizeyatu, Tadesse Sisay, Mistir Lingerew, Tilaye Matebe Yayeh, Metadel Adane.

**Funding acquisition:** Zemene Berhan, Metadel Adane.

**Investigation:** Zemene Berhan, Asmamaw Malede, Adinew Gizeyatu, Tadesse Sisay, Mistir Lingerew, Mengesha Dagne, Mesfin Gebrehiwot, Gebremariam Ketema, Kassahun Bogale, Betelhiem Eneyew, Seada Hassen, Tarikuwa Natnael, Mohammed Yenuss, Leykun Berhanu, Masresha Abebe, Gete Berihun, Birhanu Wagaye, Kebede Faris, Awoke Keleb, Ayechew Ademas, Akalu Melketsadik Woldeyohanes, Alelgne Feleke, Tilaye Matebe Yayeh, Muluken Genetu Chanie, Amare Muche, Reta Dewau, Zinabu Fentaw, Wolde Melese Ayele, Wondwosen Mebratu, Bezawit Adane, Tesfaye Birhane Tegegne, Elsabeth Addisu, Mastewal Arefaynie, Melaku Yalew, Yitayish Damtie, Bereket Kefale, Zinet Abegaz Asfaw, Atsedemariam Andualem, Belachew Tegegne, Emaway Belay, Metadel Adane.

**Methodology:** Zemene Berhan, Asmamaw Malede, Adinew Gizeyatu, Tadesse Sisay, Mistir Lingerew, Mengesha Dagne, Mesfin Gebrehiwot, Gebremariam Ketema, Kassahun Bogale, Betelhiem Eneyew, Seada Hassen, Tarikuwa Natnael, Mohammed Yenuss, Leykun Berhanu, Masresha Abebe, Gete Berihun, Birhanu Wagaye, Kebede Faris, Awoke Keleb, Ayechew Ademas, Akalu Melketsadik Woldeyohanes, Alelgne Feleke, Tilaye Matebe Yayeh, Muluken Genetu Chanie, Amare Muche, Reta Dewau, Zinabu Fentaw, Wolde Melese Ayele, Wondwosen Mebratu, Bezawit Adane, Tesfaye Birhane Tegegne, Elsabeth Addisu, Mastewal Arefaynie, Melaku Yalew, Yitayish Damtie, Bereket Kefale, Zinet Abegaz Asfaw, Atsedemariam Andualem, Belachew Tegegne, Emaway Belay, Metadel Adane.

**Project administration:** Zemene Berhan, Asmamaw Malede, Adinew Gizeyatu, Tadesse Sisay, Mistir Lingerew, Mengesha Dagne, Mesfin Gebrehiwot, Gebremariam Ketema, Kassahun Bogale, Seada Hassen, Tarikuwa Natnael, Mohammed Yenuss, Leykun Berhanu, Masresha Abebe, Gete Berihun, Birhanu Wagaye, Kebede Faris, Awoke Keleb, Ayechew Ademas, Akalu Melketsadik Woldeyohanes, Alelgne Feleke, Tilaye Matebe Yayeh, Muluken Genetu Chanie, Amare Muche, Reta Dewau, Zinabu Fentaw, Wolde Melese Ayele, Wondwosen Mebratu, Bezawit Adane, Tesfaye Birhane Tegegne, Elsabeth Addisu, Mastewal Arefaynie,

Melaku Yalew, Yitayish Damtie, Bereket Kefale, Zinet Abegaz Asfaw, Atsedemariam Andualem, Belachew Tegegne, Emaway Belay, Metadel Adane.

**Resources:** Zemene Berhan, Asmamaw Malede, Adinew Gizeyatu, Tadesse Sisay, Mistir Lingerew, Helmut Kloos, Mengesha Dagne, Mesfin Gebrehiwot, Gebremariam Ketema, Kassahun Bogale, Seada Hassen, Tarikuwa Natnael, Mohammed Yenuss, Leykun Berhanu, Masresha Abebe, Gete Berihun, Birhanu Wagaye, Kebede Faris, Awoke Keleb, Ayechew Ademas, Akalu Melketsadik Woldeyohanes, Alelgne Feleke, Tilaye Matebe Yayeh, Muluken Genetu Chanie, Amare Muche, Reta Dewau, Zinabu Fentaw, Wolde Melese Ayele, Wondwosen Mebratu, Bezawit Adane, Tesfaye Birhane Tegegne, Elsabeth Addisu, Mastewal Arefaynie, Melaku Yalew, Yitayish Damtie, Bereket Kefale, Zinet Abegaz Asfaw, Atsedemariam Andualem, Belachew Tegegne, Emaway Belay, Metadel Adane.

**Software:** Zemene Berhan, Asmamaw Malede, Adinew Gizeyatu, Tadesse Sisay, Mistir Lingerew, Mengesha Dagne, Mesfin Gebrehiwot, Gebremariam Ketema, Kassahun Bogale, Betelhiem Eneyew, Seada Hassen, Tarikuwa Natnael, Mohammed Yenuss, Leykun Berhanu, Masresha Abebe, Gete Berihun, Birhanu Wagaye, Kebede Faris, Awoke Keleb, Ayechew Ademas, Akalu Melketsadik Woldeyohanes, Alelgne Feleke, Tilaye Matebe Yayeh, Muluken Genetu Chanie, Amare Muche, Reta Dewau, Zinabu Fentaw, Wolde Melese Ayele, Wondwosen Mebratu, Bezawit Adane, Tesfaye Birhane Tegegne, Elsabeth Addisu, Mastewal Arefaynie, Melaku Yalew, Yitayish Damtie, Bereket Kefale, Zinet Abegaz Asfaw, Atsedemariam Andualem, Belachew Tegegne, Emaway Belay, Metadel Adane.

**Supervision:** Zemene Berhan, Asmamaw Malede, Adinew Gizeyatu, Tadesse Sisay, Mistir Lingerew, Helmut Kloos, Mengesha Dagne, Mesfin Gebrehiwot, Gebremariam Ketema, Kassahun Bogale, Betelhiem Eneyew, Seada Hassen, Tarikuwa Natnael, Mohammed Yenuss, Leykun Berhanu, Masresha Abebe, Gete Berihun, Birhanu Wagaye, Kebede Faris, Awoke Keleb, Ayechew Ademas, Akalu Melketsadik Woldeyohanes, Alelgne Feleke, Tilaye Matebe Yayeh, Muluken Genetu Chanie, Amare Muche, Reta Dewau, Zinabu Fentaw, Wolde Melese Ayele, Wondwosen Mebratu, Bezawit Adane, Tesfaye Birhane Tegegne, Elsabeth Addisu, Mastewal Arefaynie, Melaku Yalew, Yitayish Damtie, Bereket Kefale, Zinet Abegaz Asfaw, Atsedemariam Andualem, Belachew Tegegne, Emaway Belay, Metadel Adane.

**Validation:** Zemene Berhan, Asmamaw Malede, Adinew Gizeyatu, Tadesse Sisay, Mistir Lingerew, Helmut Kloos, Mengesha Dagne, Mesfin Gebrehiwot, Gebremariam Ketema, Kassahun Bogale, Betelhiem Eneyew, Seada Hassen, Tarikuwa Natnael, Mohammed Yenuss, Leykun Berhanu, Masresha Abebe, Gete Berihun, Birhanu Wagaye, Kebede Faris, Awoke Keleb, Ayechew Ademas, Akalu Melketsadik Woldeyohanes, Alelgne Feleke, Tilaye Matebe Yayeh, Muluken Genetu Chanie, Amare Muche, Reta Dewau, Zinabu Fentaw, Wolde Melese Ayele, Wondwosen Mebratu, Bezawit Adane, Tesfaye Birhane Tegegne, Elsabeth Addisu, Mastewal Arefaynie, Melaku Yalew, Yitayish Damtie, Bereket Kefale, Zinet Abegaz Asfaw, Atsedemariam Andualem, Belachew Tegegne, Emaway Belay, Metadel Adane.

**Visualization:** Zemene Berhan, Asmamaw Malede, Adinew Gizeyatu, Tadesse Sisay, Mistir Lingerew, Helmut Kloos, Mengesha Dagne, Mesfin Gebrehiwot, Gebremariam Ketema, Kassahun Bogale, Betelhiem Eneyew, Seada Hassen, Tarikuwa Natnael, Mohammed Yenuss, Leykun Berhanu, Masresha Abebe, Gete Berihun, Birhanu Wagaye, Kebede Faris, Awoke Keleb, Ayechew Ademas, Akalu Melketsadik Woldeyohanes, Alelgne Feleke, Tilaye Matebe Yayeh, Muluken Genetu Chanie, Amare Muche, Reta Dewau, Zinabu Fentaw, Wolde Melese Ayele, Wondwosen Mebratu, Bezawit Adane, Tesfaye Birhane Tegegne, Elsabeth Addisu, Mastewal Arefaynie, Melaku Yalew, Yitayish Damtie, Bereket Kefale, Zinet

Abegaz Asfaw, Atsedemariam Andualem, Belachew Tegegne, Emaway Belay, Metadel Adane.

**Writing – original draft:** Metadel Adane.

**Writing – review & editing:** Helmut Kloos, Metadel Adane.

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
