## [Decision Letter · Decision Letter 0]

8 Jan 2021

PONE-D-20-31546

Prevalence and associated factors of needle stick and sharp injuries among Healthcare workers in northwest Ethiopia

PLOS ONE

Dear Dr. Adane (PhD),

Thank you for submitting your manuscript to PLOS ONE. After careful consideration, we feel that it has merit but does not fully meet PLOS ONE’s publication criteria as it currently stands. Therefore, we invite you to submit a revised version of the manuscript that addresses the points raised during the review process.

Please face all suggestions and reply them. If not make the changes, please justify.

We look forward to receiving your revised manuscript.

Kind regards,

Ricardo Q. Gurgel, PhD

Academic Editor

PLOS ONE

Journal Requirements:

Reviewers' comments:

Reviewer's Responses to Questions

**Comments to the Author**

1. Is the manuscript technically sound, and do the data support the conclusions?

Reviewer #1: Partly

Reviewer #2: Yes

2. Has the statistical analysis been performed appropriately and rigorously? 

Reviewer #1: Yes

Reviewer #2: Yes

3. Have the authors made all data underlying the findings in their manuscript fully available?

Reviewer #1: Yes

Reviewer #2: Yes

4. Is the manuscript presented in an intelligible fashion and written in standard English?

Reviewer #1: Yes

Reviewer #2: Yes

5. Review Comments to the Author

Reviewer #1: The authors have made rigorous corrections based on previous reviewers' comments. However, there are still several major issues presented in the new manuscript.

1. line 110-112: In the 2007 census report, South Gondar Zone had an estimated total population of... The study was conducted in 2019, can the author provide recent data of South Gondar population?

2. line 167: operational definition of universal precaution: Avoiding contact with patients’ bodily fluids. It should be the "the practice in medicine to avoid contact with patients' bodily fluids". Please provide reference as other operational definition

3. line 222-223: How did you define patient who did not recover from the already existing NSSI? Is it patient that still have wound injury, develop hepatitis B or HIV?

4. Why did you combine internist, ophthalmologist, and other specialist in the same group with health officers who might not have the same chance of exposure to patients' bodily fluid? Why not combine with medical doctor? Please clarify the classification or reclassify as other writers do.

5. line 241-242: How did you define that universal precautions were posted in the institutions? Is it poster or education of universal precaution? Please clarify

6. Table 1. What kind of health care profession that need Master of Science degree in these sites?

7. Table 2: What does it mean by presence of universal precaution in health care setting? Can you clarify why there were health care setting that do not intend to practice universal precaution?

8. Table 2: Work department: Did surgical, medical, and obgyn department mean surgical ward, adult medical ward, and obgyn ward?

9. Table 2: Please clarify other location of sharps container in the bottom of the table.

Table 3: receive HBV prophylaxis. Is it HBV immunoglobulin or HBV vaccination before working in health care institution? Did you consider their knowledge of their own anti HBs level? Please clarify in method section.

10. Table 4: How do you define participants' knowledge about prevention of NSSI? Please clarify items and scoring used in method section.

11. Table 5: IPPS room as place of care after injury? What kind of room is this? Do you mean where they reported their injury in the first time?

12. It is interesting that 8.1% of NSSI happened while walking in the hospital. Can you explain what kind of situation in the study setting that even when walking in the hospital, someone could get NSSI.

13. Table 6: Almost 30% or health care workers did not have training on IPPS? Wasn't that part of their health care workers curriculum in medical faculty or nursing school? Do you mean special training (at least one or several days for IPPS)? Please clarify

14. abstract: prevent sleepy feeling through minimizing daily clinical activities. You can include the reasonable samples of suggestion in the discussion section. Did you mean minimizing clinical activities and adding more workers?

15. Line 360-361: Health education should emphasize dealing with feeling sleepy at work among HCWs working night shifts. Can you suggested proven method for this suggestion?

16. Line 263-264: Sixty-eight (78.2%) had sustained injury only one time in the previous 12 months and 8.0% recalled three or more injuries. Was there any one had 2 injuries in the previous 12 months?

Reviewer #2: Overall, the issue raised is relevant and the paper is well written. However, there are grammatical and writing errors which has to be corrected in addition to the following specific comments and questions.

Abstract:

• Use structured abstract form.

• Remove the following from abstract as it not your main objective.

“Of these, 46.0% reported that their injuries were moderate, superficial (33.3%) or severe (20.7%). Most injuries (41.4%) were caused by a suture needle or by a disposable syringe (27.6%).”

• Conclude your first objective using general terms by comparing it with other studies or standards. Where is your conclusion regarding associated factors?

Background:

• Move the following sentence before the last paragraph of the background section.

“Needle-stick incidents are associated with a number of different job factors, including heavy workload, working in surgical or intensive care units, insufficient work experience, and young age [3].”

Methods:

• Study setting: Rearrange the flow of writing of the study setting and remove redundancies.

• Study design and source population

Include inclusion and exclusion criteria.

• Operational definitions:

“Healthcare workers (HCWs): A healthcare professional whose activities involve contact with needles and other sharps during the course of their work in a healthcare facility [2].” Mention the types of health professionals involved in the definition.

Results:

• Generally, narrating every category of a variable is not recommended if you have tables. So, modify the narrations of socio-demographic characteristics of healthcare workers, organization-related characteristics, behavioral characteristics, and prevalence of needle stick and sharps injuries. Mention the largest or the lowest percentage as needed.

• Don’t use jargon words like most, majority, great majority and so on.

Discussion:

• Paragraph 2 and 4: Further justification is needed.

• Before the last paragraph of the discussion, write the overall implications of the study

Conclusions:

• Conclude your first objective using general terms by comparing it with other studies or standards.

References

• Reference 12 & 14: Check these references. They are similar except the year.

Reference 12: “Feleke BE. Prevalence and determinant factors for sharp injuries among Addis Ababa hospitals health professionals. Sci J Public Health. 2015;1(5).”

Reference 14: Feleke BE. Prevalence and determinant factors for sharp injuries among Addis Ababa hospitals health professionals. Sci J Public Health. 2013;1(5):189-93.

Table 2: Other than safety box: what do you mean? Mention those you found. Other in the last row of the table is not defined.

Table 6: Why you include those variables which are not significant in the final model?

Minor comments

• Line 142: Delete the phrase “for each own profession”. It is repetition.

• Line 161: Delete “study participant”.

• Line 201: Change “bivariate” by “bi-variable”

• Line 229: Change “Socio-demographic characteristics of the study participants” to (Socio-demographic characteristics of healthcare workers”.

• Line 234: Delete this sentence. It is repetition.

“One hundred twenty-four (42.0%) were nurses.”

• Line 301: Delete “of these findings to ours”

• Line 303: Delete “found by”

• Line 316: Delete “(P<0.01)”

• Table 1: Include keys for HCW, BSc. MSc. And MD

• Table 2: Include key for HCW

• Table 5: Include key for NSSI

• Table 6: Include keys for HCW, BSc, MSc, MD, and USD

6. PLOS authors have the option to publish the peer review history of their article (what does this mean?). If published, this will include your full peer review and any attached files.

Reviewer #1: No

Reviewer #2: No

---

## [Author Response · Author response to Decision Letter 0]

5 Mar 2021

Date: March 04, 2021

Manuscript ID: PONE-D-20-31546R1

Prevalence and associated factors of needle stick and sharp injuries among Healthcare workers in northwest Ethiopia

Corresponding author: Metadel Adane (PhD)

Dear Ricardo Q. Gurgel, PhD

Academic Editor

PLOS ONE

Thank you for your letter dated January 08, 2021 with a decision of revision needed. We were pleased to know that our manuscript was considered potentially acceptable for publication in PLoS ONE, subject to adequate revision as requested by the reviewers, academic editors and the journals. Based on the instructions provided in your letter, we uploaded the file of the rebuttal letter; the marked up copy of the revised manuscript highlighting the changes made in the original submitted version and the clean copy of the revised manuscript. 

We have revised the manuscript by modifying the abstract, introduction, methods, results, discussion and other sections, based on the comments made by the reviewers and using the journal guidelines. Accordingly, we have marked in red color all the changes made during the revision process. Appended to this letter is our point-by-point response (rebuttal letter) to the comments made by the reviewers. 

We agree with almost all the comments/questions raised by the reviewers and provided justification for disagreeing with some of them. We would like to take this opportunity to express our thanks to the reviewers for their valuable comments and to thank you for allowing us to resubmit a revision of the manuscript. 

I hope that the revised manuscript is accepted for publication in PLoS ONE. 

Sincerely yours,

Metadel Adane (PhD) 

Journal Requirements

Response: We formatted the manuscript using PLoS ONE format accordingly (Please see the revised version).

Reviewer #1:

The authors have made rigorous corrections based on previous reviewers' comments. However, there are still several major issues presented in the new manuscript.

Response: Many thanks for your positive reflection about the several times of revision we made on our paper. All your concerns have been addressed and please find our point by point response here below and please kindly see the revised version within the track change. We found that your comments substantially improved the manuscript and we really thank you for your commitment and scientific contribution. 

1. line 110-112: In the 2007 census report, South Gondar Zone had an estimated total population of... The study was conducted in 2019, can the author provide recent data of South Gondar population?

Response: Thank you for this key comment. Since our study was not a community-based and we found that writing the population size of the south Gondar zone is not as such relevant for our study. We noted the important information about the number of hospitals, woreads, location and etc. which is enough. Therefore, we deleted information related to population census due to the data may not be reliable as a result of not nationally accepted data by CSA (central statistical agency) of Ethiopia. We hope that the reviewer will understand our concern and justification in this matter. 

2. line 167: operational definition of universal precaution: Avoiding contact with patients’ bodily fluids. It should be the "the practice in medicine to avoid contact with patients' bodily fluids". Please provide reference as other operational definition

Response: We cited reference for the definitions of universal precaution (see in page 8 line 182). 

3. line 222-223: How did you define patient who did not recover from the already existing NSSI? Is it patient that still have wound injury, develop hepatitis B or HIV?

Response: Thank you for bringing this concern to our attention. Yes, if the patient believed that not recovered from the injuries, it is considered as a patient. However, we did not examine if that injured person develop hepatitis B or HIV and also we do know hepatitis B or HIV is even due to NSSI injury or not. 

4. Why did you combine internist, ophthalmologist, and other specialist in the same group with health officers who might not have the same chance of exposure to patients' bodily fluid? Why not combine with medical doctor? Please clarify the classification or reclassify as other writers do.

Response: This is because of the frequency of the data. In each hospital the number of internist, ophthalmologist, and other specialist is mainly one or two or three and even sometimes no in the specific category. To make the model suitable during the data analysis, we merged that very less frequent profession in the same category regardless of the exposure level. 

5. line 241-242: How did you define that universal precautions were posted in the institutions? Is it poster or education of universal precaution? Please clarify

Response: The poster for universal precaution was posted within the hospital, but we found that most (71.9%) of the health professionals did not know that universal precautions poster were posted in their institutions. We updated the manuscript by including poster. 

6. Table 1. What kind of health care profession that need Master of Science degree in these sites?

Response: No standard limitation as every health care worker can study master by private or sponsorship of the hospital. Being graduated in a master degree has no special privilege in civil service, however it will account as additional experience with salary increase compared to not graduated a master degree. Some fields such as health officer are expected to do masters in integrated emergency surgery (ISO). Thus, nurses, medical laboratory, health officers can have a master. 

7. Table 2: What does it mean by presence of universal precaution in health care setting? Can you clarify why there were health care setting that do not intend to practice universal precaution?

Response: We mean that the presence of universal precautions poster posted in your working area, there are always standard precaution measures but may not be posted in the important areas to remind the HCWs. Posters of universal precautions should not be stored rather than posted within the different areas of the HCF in a sufficient manner, so that patients and HCWs can read every time and keep themselves health. 

8. Table 2: Work department: Did surgical, medical, and obgyn department mean surgical ward, adult medical ward, and obgyn ward?

Response: Yes, we mean that as you suggested. We updated Table 2 by including ward for the different departments. Thank you. 

9. Table 2: Please clarify other location of sharps container in the bottom of the table.

Response: We explained the other locations within the Table 2 as sharps located Outside of the different wards near the door, laboratory room and pharmacy room. Thank you. (See Table 2 foot note). 

Table 3: receive HBV prophylaxis. Is it HBV immunoglobulin or HBV vaccination before working in health care institution? Did you consider their knowledge of their own anti HBs level? Please clarify in method section.

Response: Thank you for this important questions. We mean that HBV vaccination before working in health care institution. We noted this in Table 2 foot note and please see Table 2. 

10. Table 4: How do you define participants' knowledge about prevention of NSSI? Please clarify items and scoring used in method section.

Response: Thank you for this key questions. We made error by saying knowledge, but our study was either the know NSSI can be prevented or not, we updated Table 4. We did not study knowledge using different items of questions. 

11. Table 5: IPPS room as place of care after injury? What kind of room is this? Do you mean where they reported their injury in the first time?

Response: It is to mean that the existence of infection prevention and patient safety room as a care of injury place. We mean that a room that an injured HCWs reported injury in the first time for treatment. 

12. It is interesting that 8.1% of NSSI happened while walking in the hospital. Can you explain what kind of situation in the study setting that even when walking in the hospital, someone could get NSSI.

Response: Thank you for your concern. Walking in the hospital means that when going to toilet, during moving from one ward to the other ward, during moving from one location to other location during tea break time and etc. 

13. Table 6: Almost 30% or health care workers did not have training on IPPS? Wasn't that part of their health care workers curriculum in medical faculty or nursing school? Do you mean special training (at least one or several days for IPPS)? Please clarify

Response: We appreciate your comments. Taking infection prevention and patient safety training is the requirement that is recommended by the health bureau. However, the training given based on schedule and HCWs may be not be taken IPPS training during that time. Some newly recruited HCWs also may not take IPPS training during data collection. The training mostly given once but there may be also taking of the training two times or more. 

14. Abstract: prevent sleepy feeling through minimizing daily clinical activities. You can include the reasonable samples of suggestion in the discussion section. Did you mean minimizing clinical activities and adding more workers?

Response: Yes, we mean that high work load is a means for sleepy during the work place because of shortage of time for taking free time for relaxes. We updated the discussion as suggested. 

15. Line 360-361: Health education should emphasize dealing with feeling sleepy at work among HCWs working night shifts. Can you suggested proven method for this suggestion?

Response: Avoiding/minimizing feeling sleepy at work among HCWs working night and day shifts. HCWs who are in duty at night might be good if they are free at the day time and those HCWs who are at the day time duty also might be good to be free during the night in order to prevent sleepy behaviors (See in page 16). 

16. Line 263-264: Sixty-eight (78.2%) had sustained injury only one time in the previous 12 months and 8.0% recalled three or more injuries. Was there any one had 2 injuries in the previous 12 months?

Response: Yes, a total of 12 HCWs, which account 13.8% had injured two times during the last 12 months (See Table 5). 

Reviewer #2:

Overall, the issue raised is relevant and the paper is well written. However, there are grammatical and writing errors which has to be corrected in addition to the following specific comments and questions.

Response: Dear reviewer, many thanks for your recognition of our work and we have addressed all your concerns here below. 

Abstract:

• Use structured abstract form.

Response: We formatted the abstract using structured form. Thank you. 

• Remove the following from abstract as it not your main objective.

“Of these, 46.0% reported that their injuries were moderate, superficial (33.3%) or severe (20.7%). Most injuries (41.4%) were caused by a suture needle or by a disposable syringe (27.6%).”

Response: We keep this as a descriptive in the abstract because injury is our main concern. Knowing the level (types) of injury also very important for readers and intervention purpose. Furthermore, putting the associated factors from the adjusted analysis is also important. 

Conclude your first objective using general terms by comparing it with other studies or standards. Where is your conclusion regarding associated factors?

Response: We concluded that factors significantly associated with NSSIs were occupation as a nurse, habit of needle recapping, disposal of sharp materials in places other than in safety boxes and feeling sleepy at work (Please see the conclusion within the abstract and below the discussion section). Thank you for this key comments. 

Background:

• Move the following sentence before the last paragraph of the background section.

“Needle-stick incidents are associated with a number of different job factors, including heavy workload, working in surgical or intensive care units, insufficient work experience, and young age [3].”

Response: Many thanks for this view. Well accepted as it is and we moved it as suggested. 

Methods:

• Study setting: Rearrange the flow of writing of the study setting and remove redundancies.

• Study design and source population

Include inclusion and exclusion criteria.

Response: Thank you and we provided detail information as suggested. The inclusion and exclusion criteria were written within sub-title (See in page 6 from 127 to 131). 

• Operational definitions:

“Healthcare workers (HCWs): A healthcare professional whose activities involve contact with needles and other sharps during the course of their work in a healthcare facility [2].” Mention the types of health professionals involved in the definition.

Response: Thank you and we included the HCWs based on the given comment. 

Results:

• Generally, narrating every category of a variable is not recommended if you have tables. So, modify the narrations of socio-demographic characteristics of healthcare workers, organization-related characteristics, behavioral characteristics, and prevalence of needle stick and sharps injuries. Mention the largest or the lowest percentage as needed.

Response: We appreciate your comment and we fully accepted the given and amended the result section (See the result sections)

• Don’t use jargon words like most, majority, great majority and so on.

Response: We avoided. 

Discussion:

• Paragraph 2 and 4: Further justification is needed.

Response: We provided more strong justification and please see the updated version. Thank you indeed. 

• Before the last paragraph of the discussion, write the overall implications of the study

Response: We tried to write implication by giving its own sub-titles. Please see in page 17 ---- from lines 370 to 379. 

Conclusions:

• Conclude your first objective using general terms by comparing it with other studies or standards.

Response: We agree with your comments, but we faced that a standard cut of point of the prevalence of NSSIs. We could compare with other studies to see in different context, but concluding high or low by comparing with other studies looks not logical due to the fact that saying high or low needs recommended standard. 

References

• Reference 12 & 14: Check these references. They are similar except the year.

Reference 12: “Feleke BE. Prevalence and determinant factors for sharp injuries among Addis Ababa hospitals health professionals. Sci J Public Health. 2015;1(5).”

Reference 14: Feleke BE. Prevalence and determinant factors for sharp injuries among Addis Ababa hospitals health professionals. Sci J Public Health. 2013;1(5):189-93.

Response: Thank you for identifying such error, we did the revision. 

Table 2: Other than safety box: what do you mean? Mention those you found. Other in the last row of the table is not defined.

Response: Other than safety box means not using safety box. Other in the last row Table 2 means that outside of the different wards near the door, laboratory room and pharmacy room. 

Table 6: Why you include those variables which are not significant in the final model?

Response: We updated Table 6 by deleting non-significant variables. Thank you for this important comment which improved the paper very well. 

Minor comments

• Line 142: Delete the phrase “for each own profession”. It is repetition.

Response. Thank you. It is deleted 

• Line 161: Delete “study participant”.

Response: Thank you. It is deleted

• Line 201: Change “bivariate” by “bi-variable”

Response: It is changed 

• Line 229: Change “Socio-demographic characteristics of the study participants” to (Socio-demographic characteristics of healthcare workers”.

Response: It is changed 

• Line 234: Delete this sentence. It is repetition.

“One hundred twenty-four (42.0%) were nurses.”

Response: It is deleted 

• Line 301: Delete “of these findings to ours”

Response: It is deleted 

• Line 303: Delete “found by”

Response: It is deleted

• Line 316: Delete “(P<0.01)”

Response: It is deleted

• Table 1: Include keys for HCW, BSc. MSc. And MD

Response: All included 

• Table 2: Include key for HCW

Response: All included

• Table 5: Include key for NSSI

Response: Included

• Table 6: Include keys for HCW, BSc, MSc, MD, and USD

Response: Not necessary since the table updated by consisting of only significant variables in the final model. 

We would like to thank the reviewers and editors for evaluating our manuscript. We have tried to address all the concerns in a proper way and believe that our paper has been improved considerably. We would be happy to make further corrections if necessary and look forward to hearing from you all soon. 

I hope that the revised manuscript is accepted for publication in PLoS ONE. 

Sincerely yours,

Metadel Adane (PhD in Water and Public Health)

---

## [Decision Letter · Decision Letter 1]

31 Mar 2021

PONE-D-20-31546R1

Prevalence and associated factors of needle stick and sharp injuries among Healthcare workers in northwest Ethiopia

PLOS ONE

Dear Dr. Adane,

Thank you for submitting your manuscript to PLOS ONE. After careful consideration, we feel that it has merit but does not fully meet PLOS ONE’s publication criteria as it currently stands. Therefore, we invite you to submit a revised version of the manuscript that addresses the points raised during the review process.

We look forward to receiving your revised manuscript.

Kind regards,

Ricardo Q. Gurgel, PhD

Academic Editor

PLOS ONE

Journal Requirements:

Reviewers' comments:

Reviewer's Responses to Questions

**Comments to the Author**

1. If the authors have adequately addressed your comments raised in a previous round of review and you feel that this manuscript is now acceptable for publication, you may indicate that here to bypass the “Comments to the Author” section, enter your conflict of interest statement in the “Confidential to Editor” section, and submit your "Accept" recommendation.

Reviewer #1: (No Response)

Reviewer #2: All comments have been addressed

2. Is the manuscript technically sound, and do the data support the conclusions?

Reviewer #1: Partly

Reviewer #2: Yes

3. Has the statistical analysis been performed appropriately and rigorously? 

Reviewer #1: Yes

Reviewer #2: Yes

4. Have the authors made all data underlying the findings in their manuscript fully available?

Reviewer #1: Yes

Reviewer #2: Yes

5. Is the manuscript presented in an intelligible fashion and written in standard English?

Reviewer #1: Yes

Reviewer #2: Yes

6. Review Comments to the Author

Reviewer #1: Authors revised the manuscript.Some have been addressed, but some modifications still required.

1. line 167: operational definition of universal precaution: Avoiding contact with patients’ bodily fluids. It should be the "the practice in medicine to avoid contact with patients' bodily fluids". Please provide reference as other operational definition

Response: We cited reference for the definitions of universal precaution (see in page 8 line 182).

Comments: As suggested before, the author need to use more standard definition and cite from appropriate reference. The citated reference is not easy to find and wrongly written.

2. Why did you combine internist, ophthalmologist, and other specialist in the same group with health officers who might not have the same chance of exposure to patients' bodily fluid? Why not combine with medical doctor? Please clarify the classification or reclassify as other writers do.

Response: This is because of the frequency of the data. In each hospital the number of internist, ophthalmologist, and other specialist is mainly one or two or three and even sometimes no in the specific category. To make the model suitable during the data analysis, we merged that very less frequent profession in the same category regardless of the exposure level.

Comments: I still found the reason of combining these specialists with health officers not reasonable. Especially, it was just based on suitable data analysis.

3. Table 3: receive HBV prophylaxis. Is it HBV immunoglobulin or HBV vaccination before working in health care institution? Did you consider their knowledge of their own anti HBs level? Please clarify in method section.

Response: Thank you for this important questions. We mean that HBV vaccination before working in health care institution. We noted this in Table 2 foot note and please see Table 2.

Comment: It is better to call as HBV vaccination (completed or not-completed). I assumed the author did not have data on HBV vaccine completion.

4. Table 4: How do you define participants' knowledge about prevention of NSSI? Please clarify items and scoring used in method section.

Response: Thank you for this key questions. We made error by saying knowledge, but our study was either the know NSSI can be prevented or not, we updated Table 4. We did not study knowledge using different items of questions.

Comment: Does it mean prevention NSSI or prevention of infection caused by NSSI? These are two different terms

5. It is interesting that 8.1% of NSSI happened while walking in the hospital. Can you explain what kind of situation in the study setting that even when walking in the hospital, someone could get NSSI.

Response: Thank you for your concern. Walking in the hospital means that when going to toilet, during moving from one ward to the other ward, during moving from one location to other location during tea break time and etc.

Comment: If this was the situation in that hospital, I would suggest the author describe with more data and emphasize more in the discussion. Please also include realistic suggestion to this horrible situation.

6. It is interesting that 8.1% of NSSI happened while walking in the hospital. Can you explain what kind of situation in the study setting that even when walking in the hospital, someone could get NSSI.

Response: Thank you for your concern. Walking in the hospital means that when going to toilet, during moving from one ward to the other ward, during moving from one location to other location during tea break time and etc.

Comment: If this was the situation in that hospital, I would suggest the author describe with more data and emphasize more in the discussion. Please also include realistic suggestion to this horrible situation.

Reviewer #2: The authors have improved the paper very well. All the comments are properly addressed.

•The abstract is well structured,

•The background section is rearranged,

•The methodological issues are corrected,

•The write up of the results, discussion and conclusion section are improved,

•The implications of the study are indicated,

•Corrections are properly made on references,

•Writing and grammatical errors are corrected and

•Key notes of tables are properly defined and written.

By now, I will be happy if the paper is published at PLOS ONE.

7. PLOS authors have the option to publish the peer review history of their article (what does this mean?). If published, this will include your full peer review and any attached files.

Reviewer #1: **Yes: **Evy Yunihastuti, MD, PhD

Reviewer #2: **Yes: **Getaw Walle Bazie

---

## [Author Response · Author response to Decision Letter 1]

12 Apr 2021

Response to reviewers 

Reviewer #1: Authors revised the manuscript. Some have been addressed, but some modifications still required.

Response: Thank you for your positive remark for our revision. We revised the manuscript as per your comments and please see the answers herewith. 

1. Line 167: operational definition of universal precaution: Avoiding contact with patients’ bodily fluids. It should be the "the practice in medicine to avoid contact with patients' bodily fluids". Please provide reference as other operational definition. 

Response: We cited reference for the definitions of universal precaution (see in page 8 line 182).

Comments: As suggested before, the author need to use more standard definition and cite from appropriate reference. The citied reference is not easy to find and wrongly written.

Response: We really appreciate your important comment. We updated the definitions and the reference based on your comment. Please see the revised version under the operational definitions in page 8. 

2. Why did you combine internist, ophthalmologist, and other specialist in the same group with health officers who might not have the same chance of exposure to patients' bodily fluid? Why not combine with medical doctor? Please clarify the classification or reclassify as other writers do.

Response: This is because of the frequency of the data. In each hospital the number of internist, ophthalmologist, and other specialist is mainly one or two or three and even sometimes no in the specific category. To make the model suitable during the data analysis, we merged that very less frequent profession in the same category regardless of the exposure level.

Comments: I still found the reason of combining these specialists with health officers not reasonable. Especially, it was just based on suitable data analysis.

Response: Thank you for your reasonable questions. I have to brief about the role and responsibilities of health officers in Ethiopian healthcare context. Health officers in Ethiopia are doing the same job as physicians where there was no physicians in the hospital. Moreover, even when there is physicians, health officers is do similar job and the various of activities depends on the level or status of the cases (patients) degree of severity. In all these circumstance, the exposure status is similar. Therefore, merging of our data is reasonable. 

3. Table 3: receive HBV prophylaxis. Is it HBV immunoglobulin or HBV vaccination before working in health care institution? Did you consider their knowledge of their own anti HBs level? Please clarify in method section.

Response: Thank you for this important questions. We mean that HBV vaccination before working in health care institution. We noted this in Table 2 foot note and please see Table 2.

Comment: It is better to call as HBV vaccination (completed or not-completed). I assumed the author did not have data on HBV vaccine completion.

Response: Yes, as you noted very well, the limitation of our data did not consist about the completed and not-completed HBV vaccination. We are sorry for missing such data and hoping that this will not be a major problem for this paper. 

4. Table 4: How do you define participants' knowledge about prevention of NSSI? Please clarify items and scoring used in method section.

Response: Thank you for this key questions. We made error by saying knowledge, but our study was either the know NSSI can be prevented or not, we updated Table 4. We did not study knowledge using different items of questions.

Comment: Does it mean prevention NSSI or prevention of infection caused by NSSI? These are two different terms

Response: Thank you for your insight. Prevention NSSI and prevention of infection caused by NSSI are two different ideas. However, in our study context, we only studied as NSSI can be prevented or not, whereas we did not study about prevention of infection caused by NSSI. 

5. It is interesting that 8.1% of NSSI happened while walking in the hospital. Can you explain what kind of situation in the study setting that even when walking in the hospital, someone could get NSSI.

Response: Thank you for your concern. Walking in the hospital means that when going to toilet, during moving from one ward to the other ward, during moving from one location to other location during tea break time and etc.

Comment: If this was the situation in that hospital, I would suggest the author describe with more data and emphasize more in the discussion. Please also include realistic suggestion to this horrible situation.

Response: Many thanks for this valuable comment. We included this in the discussion. About 8.1% of NSSI happened while walking in the hospital. This might be due to while walking in the hospital during visiting toilet, during moving from one ward to the other ward, during moving from one location to other location during tea break time. This also might have been as a result of lack of cleanness of the hospital compound from sharp materials. In addition to this, lack of monitoring of the standard infection prevention and control strategies may bring this horrible situation (See the revised version of the discussion in page 15 from lines 342-347. 

Reviewer #2: The authors have improved the paper very well. All the comments are properly addressed.

•The abstract is well structured,

•The background section is rearranged,

•The methodological issues are corrected,

•The write up of the results, discussion and conclusion section are improved,

•The implications of the study are indicated,

•Corrections are properly made on references,

•Writing and grammatical errors are corrected and

•Key notes of tables are properly defined and written.

By now, I will be happy if the paper is published at PLOS ONE.

Response: Dear reviewers, we really appreciate your recognition for our manuscript. Thank you very much indeed.

---

## [Editor Report · Decision Letter 2]

10 May 2021

Prevalence and associated factors of needle stick and sharp injuries among Healthcare workers in northwest Ethiopia

PONE-D-20-31546R2

Dear Dr. Adane,

We’re pleased to inform you that your manuscript has been judged scientifically suitable for publication and will be formally accepted for publication once it meets all outstanding technical requirements.

Kind regards,

Ricardo Q. Gurgel, PhD

Academic Editor

PLOS ONE
---

## [Editor Report · Acceptance letter]

17 Sep 2021

PONE-D-20-31546R2 

Prevalence and associated factors of needle stick and sharps injuries among healthcare workers in northwestern Ethiopia 

Dear Dr. Adane:

I'm pleased to inform you that your manuscript has been deemed suitable for publication in PLOS ONE. Congratulations! Your manuscript is now with our production department. 

Kind regards, 

on behalf of

Professor Ricardo Q. Gurgel 

Academic Editor

PLOS ONE